# The Potential Role of Epigallocatechin-3-Gallate (EGCG) in Breast Cancer Treatment

**DOI:** 10.3390/ijms241310737

**Published:** 2023-06-27

**Authors:** Víctor Marín, Viviana Burgos, Rebeca Pérez, Durvanei Augusto Maria, Paulo Pardi, Cristian Paz

**Affiliations:** 1Laboratory of Natural Products & Drug Discovery, Center CEBIM, Department of Basic Sciences, Faculty of Medicine, Universidad de La Frontera, Temuco 4780000, Chile; 2Departamento de Ciencias Biológicas y Químicas, Facultad de Recursos Naturales, Universidad Católica de Temuco, Rudecindo Ortega, Temuco 02950, Chile; 3Departamento de Ciencias Básicas, Facultad de Ciencias, Universidad Santo Tomas, Temuco 4780000, Chile; 4Development and Innovation Laboratory, Butantan Institute, Sao Paulo 1500, Brazil; 5Nucleo de Pesquisas NUPE/ENIAC University Center, Guarulhos 07012-030, Brazil

**Keywords:** epigallocatechin-3-gallate, breast cancer, synergistic chemotherapy, metabolism, drug delivery

## Abstract

Breast cancer is one of the most diagnosed cancers worldwide, with an incidence of 47.8%. Its treatment includes surgery, radiotherapy, chemotherapy, and antibodies giving a mortality of 13.6%. Breast tumor development is driven by a variety of signaling pathways with high heterogeneity of surface receptors, which makes treatment difficult. Epigallocatechin-3-gallate (EGCG) is a natural polyphenol isolated as the main component in green tea; it has shown multiple beneficial effects in breast cancer, controlling proliferation, invasion, apoptosis, inflammation, and demethylation of DNA. These properties were proved in vitro and in vivo together with synergistic effects in combination with traditional chemotherapy, increasing the effectiveness of the treatment. This review focuses on the effects of EGCG on the functional capabilities acquired by breast tumor cells during its multistep development, the molecular and signal pathways involved, the synergistic effects in combination with current drugs, and how nanomaterials can improve its bioavailability on breast cancer treatment.

## 1. Introduction

Breast cancer is one of the most frequent malignancies diagnosed in women around the world, and at the same time, it is the leading cause of cancer death in adult women [1]. It is a widely heterogeneous disease, especially in its advanced stages [2]. It can be classified into phenotypes depending on the cellular expression of three receptors: the estrogen receptor (ER), the progesterone receptor (PR), and receptor 2 of the epidermal human growth factor (HER2 or ERBB2). It is defined as triple-negative breast cancer (TNBC), a phenotype characterized by the absence of the three receptors [3]; in the clinic, this phenotype makes up between 15 and 20% of all breast cancers diagnosed today. These types of cancers have an aggressive course, presenting a greater capacity to metastasize and thus worsen the prognosis [4]. TBNC patients cannot benefit from targeted anti-receptor or hormone replacement therapies, leaving only systemic and highly cytotoxic non-targeted treatments as an option [5]. Chemotherapy for breast cancers is based on the anthracycline–taxane association, with a median overall survival of 36 months [6]. Furthermore, TNBCs are remarkably heterogeneous at the transcriptional level, making their treatment and prognosis even more difficult [7]. In this way, TNBC therapy includes chemotherapy, radiotherapy, and surgery, being common to observe a high resistance to drugs in the tumor mass, the appearance of strong side effects, a high tendency to relapse, and a high mortality rate in these patients [3].

Natural compounds have gained interest in recent times as adjuvants in new therapeutic strategies to treat a variety of cancers and provide new alternatives to improve the treatment of patients who cannot use targeted and less aggressive therapies [5]. One of these natural products is green tea, a derivative of the leaves of the *Camellia sinensis* plant; it is one of the most consumed beverages worldwide, and its use has spread over thousands of years in various oriental cultures, such as China and India [8]. The flavonoids present in its leaves, mainly catechins, represent 20 to 30% of its dry weight [9]. The main catechins are epigallocatechin-3-gallate (EGCG), epigallocatechin (EGC), epicatechin (EC), and epicatechin gallate (ECG) [10], as shown in Figure 1.

The catechins show diverse anti-inflammatory, anticarcinogenic, metal chelating, radical scavenging, and other biological activities [11,12]. Epidemiological studies have described a certain trend between the cultural consumption of green tea and the low prevalence of breast cancer in some populations [13]. Epigallocatechin-3-gallate (EGCG) is the most abundant polyphenol in green tea, and it has been shown in vitro and in vivo to act by modulating various signaling pathways through which it can exert a protective anticarcinogenic effect. In addition, EGCG, in combination with natural or synthetic drugs, improved the response to overcome the highly deleterious effects of non-targeted therapies.

## 2. Protective Mechanisms of EGCG in Breast Cancer

The clinical approach to breast cancer is complex due to the breadth and scope of its diversity, encompassing genetics, cell and tissue biology, pathology, and response to therapy. EGCG has shown multiple effects on the major signaling pathways governing carcinogenesis and cancer progression, including MAP kinase (MAPK), phosphatidylinositol-3 kinase (PI3K), nuclear factor κB (NFkB), and reducing the increased levels of phosphorylation of ERK1/2 and Jak/STAT3 [14]. In addition, it acts by modulating the receptors typically expressed in breast cancer, such as ER and ErbB [15,16,17].

### 2.1. Suppressive Effects of EGCG on Proliferative Signals

The signaling of the ER, PR, and HER2 receptors are overexpressed in the different cell phenotypes of breast cancer; in this sense, EGCG showed positive effects, decreasing their activities. For example, in MCF-7 and MDA-MB-231 cell lines, EGCG showed a similar cytotoxic effect independent of the activation of ER and PR receptors [18]. In the T-47D cell line (which represents the luminal A hormone-dependent subtype of breast cancer), the incubation of EGCG and estradiol (E2) decreased the estrogen receptor alpha (Erα), which promotes cell division in the presence of estrogen (17β-estradiol). Therefore, EGCG can be considered a phytoestrogen due to its structural similarity to E2, a characteristic that allows it to generate estrogenic or anti-estrogenic effects after binding to Erα receptors [18], inhibiting the growth of several ER-negative breast cancer stem cells, such as SUM-149, SUM-190, and MDA-MB-231, in which ER-α receptor variants are expressed such as ER-α36, causing a reduction in the expression of such receptors. EGCG inhibits the proliferation of triple-negative cells, MDA-MB-231 and MDA-MB-436, by the inhibition of ER-α36 [19].

EGCG inhibits the expression of epidermal growth factor receptors (EGFR or ErbB) such as ErbB1 and ErbB2, which are overexpressed in breast cancer [19], especially in epidermoid carcinoma (A-431) and SK-BR3. Moreover, it causes a reduction in cell viability by mitochondrial collapse, increased production of reactive oxygen species (ROS), and changes in nuclear morphology [20]. The ErbB receptor family comprises four paralogous receptor tyrosine kinases, EGFR (ErbB1/HER1), ErbB2 (HER2), ErbB3 (HER3), and ErbB4 (HER4), which are activated by a variety of growth factors, including the epidermal growth factor (EGF), betacellulin (BTC), transforming growth factor alpha (TGF-α), and neuregulin 1 (NRG1). ErbB receptors share a common structure that is characterized by extracellular ligand binding and intracellular kinase domains, which are separated by a single-pass transmembrane domain [21].

ErbB receptors organize into heterodimers after their conformation is induced by ligands, thus promoting transmembrane signaling, which is overexpressed in cancer cells [22]. This dysregulation stimulates proliferation via the PI3K/Akt pathways and the Ras/Raf/mitogen-activated protein kinase (MAPK) pathway, which stimulates the activation of proliferation and survival gene transcription factors. EGCG has demonstrated decreasing effects on the expression of these receptors due to changes in the organization of the lipidic rafts in the plasma membrane, interfering with their binding to EGF, or due to an increase in consumption through internalized endosomes [23,24]. ECGC makes TNBC cells sensitive to estrogen via activation of ER-α and CCN5/WISP-2 expression, inducing apoptosis and thus reducing proliferative effects in MCF-7 (ER-α positive), MDA-MB-231, and HCC-70 (TNBC) line cells [25]. Moreover, EGCG inhibits the expression of the progesterone receptor isoforms (PR-A/B proteins), suggesting an effect on cell viability by both ER and PR receptors [26].

### 2.2. EGCG Inhibits Evasion of Apoptosis in Cancer

EGCG promotes the inhibition of the PI3K/AKT pathways in T47D cells by gene and protein expression [27], accompanied by a promotion of pro-apoptotic genes such as p53, p21, cas3, cas9, Bax, and PTEN and a reduction of anti-apoptotic survival genes, such as PI3K, AKT, and Bcl-2, increasing the protein expression ratio Bax/Bcl-2. The pro-apoptotic effects of EGCG are also related to a decrease in the gene expression of hTERT, the catalytic subunit of the telomerase enzyme, which stimulates the induction of cell senescence [27]. hTERT overexpression can predict the survival of cancer and is associated with TNM stage, lymphatic metastasis, and a poor prognosis [28].

EGCG is also capable of reducing the expression of STAT3-NFkB by inhibiting the selective phosphorylation of STAT3 and its subsequent translocation to the nucleus, thus preventing its interaction with the NFkB factor and negatively regulating the expression of CD44 [29]. The STAT3 factor has been found to be aberrantly expressed in cancer stem cells (CSC) of breast tumors, being important in formation, self-renewal, and differentiation [27], promoting oncogenes such as c-MYC, S-phase associated protein kinase 2 (SKP2), and cyclin D1, as well as anti-apoptotic proteins such as Bcl-2, Bcl-xL, Mcl-1, and survivin [30]. In this way, the signaling inhibition of NFkB by STAT3 means that NFkB cannot stimulate anti-apoptotic genes and suppress pro-apoptotic ones, as usually happens during cancer progression [29].

### 2.3. EGCG Controls the Replicative Potential of Cancer

EGCG modulates the β-catenin pathway, which is dependent on Wnt signals and is usually overexpressed in cancer cells, which promotes the dephosphorylation of β-catenin, its accumulation, and subsequent translocation to the cell nucleus, where it acts as a transcription factor for cell fate, differentiation, proliferation, and stem cell pluripotency. These signals are critical in normal breast development, where EGCG has been shown to inhibit the canonical pathway of this signaling in MDA-MB-231 and MCF-7 cell lines [31,32,33] by a partial deactivation or dephosphorylation of the Akt proteins, essential for cell metabolism, growth, and survival [33]. Additionally, it has been demonstrated to modulate the CCND1 gene (cyclin D1 gene), an important factor for cell cycle progression and G1/S phase transition with antiproliferative effects [33,34].

### 2.4. EGCG Inhibits Tissue Invasion and Metastasis

EGCG reduces Golgi Membrane Protein 1 (GOLM1) expression in MDA-MB-231 cells, which is overexpressed in many solid tumors, through the HGF/HGFR/AKT/GSK-3/β-catenin/c-Myc signaling pathway. GOLM1 is a protein that increases tumor growth and metastasis, promoting the migration effects of cancer cells [35]. The anti-invasive effect elicited by EGCG was demonstrated by the inhibition of matrix metalloproteinase 2 and 9 (MMP-2 and MMP-9) activities in multidrug-resistant (MDR) human breast adenocarcinoma cells (CF7/DOX) [36]. MMP-2 and MMP-9 are part of a key group of proteolytic enzymes in the degradation of the extracellular matrix in invasive processes, together with the proteins cathepsins and plasminogen, and constitute one of the key factors in the metastatic potential and the invasive capacity of new organs by tumor cells [37].

Additionally, EGCG-induced dephosphorylation of kinase proteins in the Focal Adhesion Kick (FAK) pathway, which encodes protein tyrosine kinases that are normally concentrated in focal adhesions formed between growing cells in the presence of extracellular matrix components, suggests that EGCG inhibits the cell adhesion capacity in cancer cell propagation [34]. Figure 2 shows the multiple activities of EGCG on breast cancer signaling.

### 2.5. Effects of EGCG on the Immune System

Myeloid-derived suppressor cells (MDSCs) are responsible for the typical immunosuppression of breast cancer cells, which is inhibited by EGCG in 4T1 cells [38]. MDSC-mediated immunosuppression is widely described in human and mouse tumors and is characterized by the consequent activation of immune suppression factors such as Arg-1 (arginase 1), iNOS (inducible nitric oxide synthase), NO (nitric oxide), and ROS (reactive oxygen species) [39,40]. In humans, MDSCs are classified into monocytic-MDSCs (CD11b+ CD14+ HLA-DR−/low CD15−) and granulocytic-MDSCs (CD11b+ CD14− HLA-DR-/low CD15+). In breast cancer patients, MDSCs are functionally and phenotypically similar to bone marrow-derived MDSCs, suggesting their origin in the precursors of bone marrow [41].

EGCG suppressed the cell viability, migration, and invasion of 4T1 cells and also induced apoptosis at concentrations above 50 µg/mL. At lower concentrations (5 µg/mL), EGCG arrested MDSCs at G0/G1phases and promoted apoptosis in 4T1. The inhibition of MDSCs by EGCG was evidenced for downregulation of the canonical regulation axis Arg-1/iNOS/Nox2/NF-κB/STAT3, reducing Nox2, p47-phox, and gp91-phox NADPH oxidase subunit expression. Moreover, NF-κB was downregulated together with cytokine expressions IL-6, IL-10, TGF-β, and GM-CSF [38]. In mice plasma, EGCG concentration rises to 0.5 µg/mL, increasing CD4+ and CD8+ in peripheral blood, spleen, and tumor tissues, and reducing MDSCs. Therefore, tumors in mice are more susceptible to immune response at the initiation, growth, and metastasis stages, providing a better perspective of therapy [38].

### 2.6. Epigenetic Regulation of Cancer

EGCG has been shown to exert an epigenetic protective mechanism in various types of cancer, such as colorectal, endometrial, and breast cancer. The tumor suppressor gene SCUBE2 is hypermethylated in breast cancer, and EGCG can block its methylation by reducing DNA methyltransferase (DNMT) activity, increasing the protein expression of E-cadherin, and decreasing vimentin [42]. Inhibition of DNMT’s activity leads to the sensitization of cancer cells to therapies by disabling or reversing methylation in certain tumor suppressor genes [43]. In this way, EGCG contributes to suppressing the processes of cell proliferation, migration, and invasion by allowing the regulation of genes involved in epithelial–mesenchymal transition (EMT) by SCUBE2. The mechanism of action of EGCG on DNMT is in the MTAsa domain of the enzyme, competing with its intrinsic inhibitor S-adenosyl-L-homocysteine (SAH). Moreover, EGCG reduces IFI16 expression and ARNm of DNMT in cancer cell lines [44]. A similar process of demethylation was demonstrated in the RASSF1A tumor suppressor gene in BT-549 human breast ductal carcinoma cells, which is usually hypermethylated. EGCG causes cell cycle arrest by preventing cyclin accumulation D1 [45].

Combinations of EGCG with other active demethylating compounds of botanical origins, such as sulforaphane (SFN), show an epigenetic protective effect, correcting the epigenetic aberrations that condition the non-expression of ERα in negative cell lines for these receptors, triggering a substantial improvement in targeted hormonal therapies, and sensitizing tumor cells to treatments based on estrogen antagonists such as tamoxifen [46]. At the post-transcriptional level, EGCG regulates gene expression mediated by short non-coding RNA sequences such as miR-25 [47]. This sequence constitutes a key regulator in cancer progression and tumor development that is overexpressed in breast cancer. In MCF-7 cells, EGCG reduces cell proliferation by inducing apoptosis after miR-25 silencing, affecting invasiveness [47].

Moreover, EGCG decreases rRNA transcription and cell proliferation in MCF-7 cells by lysine-demethylase 2A (KDM2A) activation due to AMPK activation and ROS production [48].

### 2.7. EGCG on Cancer Metabolism

EGCG interfered with glycolytic processes, decreasing the expression of key enzymes involved in the regulation of glucose metabolism and lactate production, such as phosphofructokinase (PFK), lactate dehydrogenase (LDH), and hexokinase (HK), as well as the glucose transporter GLUT1, producing lower glucose consumption, lactate production, and ATP generated by metabolism [49]. EGCG also inhibited the expression of hypoxia-inducible factor 1α (HIF1α), which is overexpressed in tumors [50]. Moreover, EGCG inhibits protein metabolism, reducing proline dehydrogenase (PRODH) overexpressed in TNBC HS578T cells, 6.6-fold more than normal cells, preventing epithelial–mesenchymal transition and metastasis in breast cancer cells, in a patient-derived xenograft (PDX) mouse model [51]. EGCG reduces cell viability by reducing the activity of the protein tyrosine phosphatase family (PTPs) as PTP1B [52], which is overexpressed in tumors, including breast cancer [53]. Moreover, EGCG interferes with leptin-induced cell proliferation, which acts as a hormone secreted by white adipocytes and other tissues such as the mammary epithelium and the placenta and regulates various bodily functions such as the maintenance of homeostasis, food intake, immunity, and metabolism [54]. In breast cancer, leptin receptors are overexpressed on the cell surface of the mammary epithelium, conditioning a greater progression of the tumor dependent on this hormone [55]. EGCG inhibit the induction of adipogenic biomarkers such as lipoprotein lipase, adiponectin, leptin, fatty acid synthase, and fatty acid binding protein 4, affecting adipose-derived mesenchymal stem cell differentiation into adipocytes and paracrine regulation of the TNBC invasive phenotype, which is correlated with increased STAT3 phosphorylation status [56]. Additionally, EGCG inhibits the signals that trigger the cancer-associated adipocyte (CAA)-like phenotype by human adipose-derived mesenchymal stem/stromal cells (ADMSC), in studies where ADMSC were exposed to human TNBC-derived MDA-MB-231 (pro-inflammatory microenvironment), reducing expression of cytokines such as CCL2, CCL5, IL-1β, and IL-6 [57].

## 3. Synergistic Effect of EGCG in the Treatment of Breast Cancer

Breast cancer monotherapies are not completely effective for the treatment of advanced neoplasms, but the combination of traditional chemotherapeutic agents together with EGCG has shown sensitization of tumor cells, improving treatment response. Here, we summarize the last combination treatments using EGCG in Table 1.

## 4. Absorption and Metabolism of EGCG

The bioavailability of EGCG was studied by Nakagawa and Miyazawa in 1997, determining that EGCG is absorbed in the intestinal mucosa. After 1 h of supply of a single dose of EGCG (500 mg/kg, rat), the concentration rose to 12.3 nmol/mL in plasma, 48.4 nmol/g in the liver, 0.5 nmol/g in the brain, 565 nmol/g in the small intestinal mucosa, and 68.6 nmol/g in the colon mucosa. The concentration of EGCG in tissue corresponds to 0.0003–0.45% of the ingested compound [70], evidencing poor bioavailability. In humans, the concentration in plasma was 857 ng·h/mL after an intake of 95 mg of EGCG and this concentration increased when it was co-administrated with caffeine 40 mg, reaching a maximum of 1370 ng·h/mL [70]. The absorption of EGCG started after 8 h, reaching a maximum concentration after 24 h of intake, suggesting that its absorption undergoes gut microbiota metabolism [71]. Bacteria of the gut rat microbiota include *Enterobacter aerogenes*, *Raoultella planticola*, *Klebsiella pneumoniae* susp. *pneumoniae*, and *Bifidobacterium longum* subsp. *Infantis* promote the hydrolysis of EGCG to EGC and gallic acid, and from these compounds, a series of metabolites are produced, which are determined in cecal, feces, and urine [72], Figure 3.

From the hydrolysis of EGCG, EGC and gallic acid are produced. Further reductive cleavage of EGC produces compound **3**; then, dehydroxylation produces compound **4**, followed by the degradation of the phloroglucinol ring to produce the main compound **5**. When EGCG was administered orally, compounds **4**, **5**, **6**, **7,** and **8** were found in thefeces. When it was administered directly to the cecum, compounds **3**, **4**, **5,** and **6** were produced. After the degradation of EGCG by gut bacteria, the main compound isolated in urine was **9** [71].

## 5. Strategies for EGCG Delivery

Despite the broad spectrum of action and innocuousness of EGCG, this is a sensitive molecule to oxidation or hydrolysis, depending on pH, temperature, and concentration, which can occur in culture media, the stomach, or the intestine [73]. Moreover, poor absorption and high degradation by gut microbiota reduce the possibilities for clinical use of this compound. These drawbacks are being overcome with the use of drug delivery technologies that load the compound in nanoethosomes, vesicle systems, nanopores, emulsions, etc. These materials show enhanced physicochemical and bioactivity properties, and the incorporation of tumor-specific markers allows increased delivery to tumors. These formulations have been proven effective against many diseases. Table 2 summarizes EGCG formulations and nanomaterials used in breast cancer studies.

## 6. Conclusions

EGCG was shown to inhibit breast cancer tumors and enhance the effects of anticancer drugs in vitro and in vivo. This transversality in its effects could be taken advantage of, especially in times when studies for the establishment of personalized therapy and the selection of alternative therapeutic candidates are increasingly prevalent. The bioavailability of EGCG is improved by nanomaterials, which can include specific cellular signals that increase the drug’s delivery to tumors. Moreover, clinical trials showed that high doses of EGCG, or green tea extract, are safe and modulate beneficial health factors. There is no doubt that EGCG constitutes a complementary and versatile adjuvant candidate for the treatment of breast cancer.

## Figures and Tables

**Figure 1 ijms-24-10737-f001:**
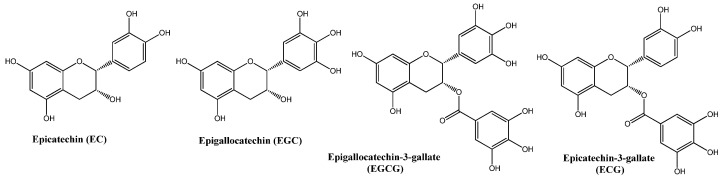
Molecular structures of main catechins identified in green tea.

**Figure 2 ijms-24-10737-f002:**
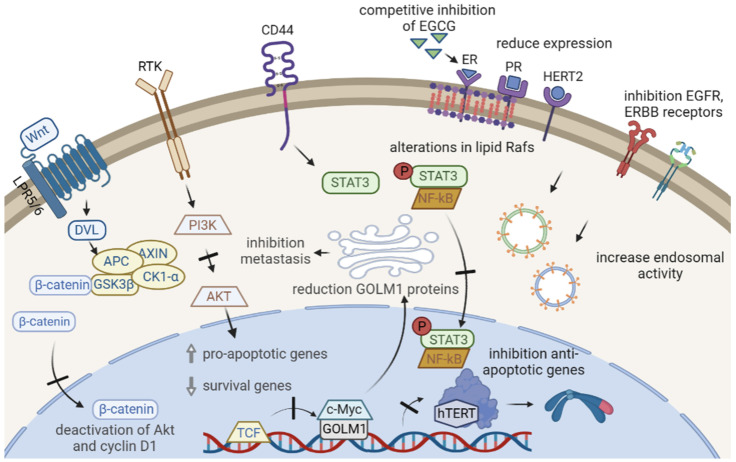
Multiple effects of EGCG in breast cancer. EGCG inhibits phosphorylation and activation of signaling pathways such as β-catenin, PI3K/AKT, and STAT3, preventing the translocation of its effectors to the nucleus. The main effect of this is the promotion of apoptosis and the inhibition of anti-apoptotic genes. In addition, this causes a decrease in the gene expression of the catalytic subunit of the telomerase enzyme hTERT. EGCG can act as a competitive inhibitor of hormones such as estradiol at key receptors for tumor growth, decrease its surface expression, alter its dispositions on the surface due to changes in lipid Rafts, or increase endosomal activity in the cell.

**Figure 3 ijms-24-10737-f003:**
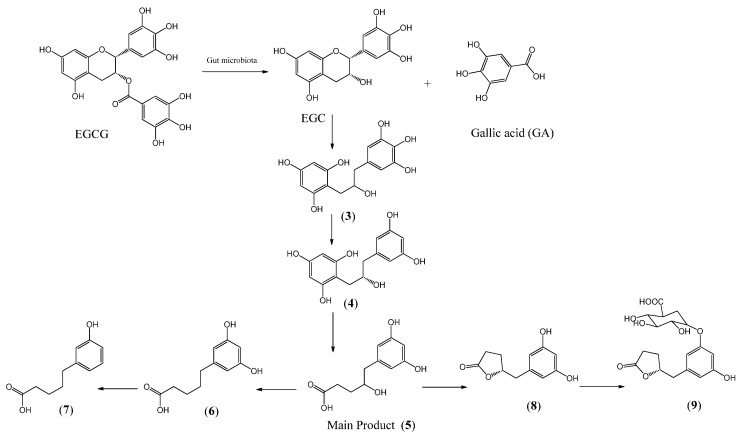
Metabolic pathways of EGCG by gut microbiota.

**Table 1 ijms-24-10737-t001:** Synergic effects of EGCG in breast cancer models.

Study	Treatment	Models	Outcomes	References
		in vitro studies		
Cell viability by cell proliferation assay	4-hydroxytamoxifen (1 μM) + 5–25 μM EGCG. 7 days	MCF-7, 47D, MDA-MB-231 and HS578T	Decrease cell viability. Synergistic activity of EGCG in MDA-MB-231 cells at 25 μM.	[58]
Cell growth, apoptosis, and epigenetic regulation of transcriptional activity of DNA	Clofarabine/ EGCG 10 μM. 4 days	MCF7 and MDA-MB-231	Decrease cell growth with low toxicity. Increase in apoptosis and reactivation of DNA methylation-silenced tumor suppressor genes such as RARB.	[59]
Effects of Tapentadol on viability and migration	Tapentadol 1–80 μg/mL + EGCG 1–160 μM	MDA-MB-231	Reduction of proliferation by impairing cell-cycle progression. Increase in apoptosis.	[60]
Inhibition insulin receptor substrate (IRS)/MAPK	24 h of sunitinib and then 12 h with pulsed EGCG 0–50 μM	MCF-7, H460, and H1975, with PIK3CA mutations	Downregulation of insulin receptor substrate (IRS), suppressed mitogenic effects, and inhibition of IRS/MAPK/p-S6K1 signaling.	[61]
Reactivation of ERa by EGCG and histone deacetylase inhibitor	Trichostatin A (TSA) (100 ng/mL for 12 h) + 10 μM EGCG	MDA-MB-231	Sensibilization of ERα-negative breast cancer cells to the activator 17β-estradiol (E2) and antagonist tamoxifen.	[62]
Effects of p53 gene silencing in conjunction with EGCG.	p53 siRNA 40 nmol + EGCG 24 h	Hs578T	Activation of pro-apoptotic and inhibition of anti-apoptotic genes, autophagy, and cell network formation.	[63]
Effects of EGCG and IIF in the EGFR inhibition	IIF 15 or 30 μM + 25 μg/mL EGCG by 24 h	MCF-7 and MDA-MB-231	Inhibition of EGFR phosphorylation, invasion, and metastasis.	[64]
Synergism of SAHA and EGCG in TNBC cells	Suberoylanilide hydroxamic acid (SAHA) 25 mM + EGCG 100 mM, every 24 h/3 days	MDA-MB-157, MDA-MB-231, and HCC1806	Increase apoptosis by decreasing cIAP2 and increasing pro-apoptotic caspase 7. Inhibition of cell migration.	[65]
Synergism of 5-aza 2’dC with EGCG	5-aza 2′dC 5 µM+ EGCG 50 µM. 7 days	MCF-7, MDA-MB 231 and MCF-10A	Synergic effects in cell growth inhibition by epigenetic mechanisms.	[66]
Evaluation of epigenetic induction of matrix metalloproteinase-3	green tea polyphenols 10 µg/mL + EGCG 20 µM 24 h	MCF-7 and MDA-MB-231	Activation of TIMP-3 and reduction of zeste homolog 2 (EZH2) and class I histone deacetylases (HDACs).	[67]
in vivo studies				
FLuc2 fusion with N-terminal 100-aa of Nrf2 and activation of Nrf2-ARE signaling	Oral cisplatin 5 mg/kg + EGCG 100 mg/kg. 11 days	MDA-MB231 tumor xenografts	Synergistic activity in vivo by Tumor size reduction in TNBC tumor xenografts.	[68]
Effects of EGCG on oral bioavailability of Tamoxifen	Tamoxifen (IV, 2 mg/kg and PO, 10 mg/kg), followed by EGCG (0.5, 3 and 10 mg/kg).	Male Sprague Dawley rats	Increase of bioavailability 1.48–1.77-fold of Tamoxifen in the presence of EGCG.	[69]
Evaluate angiogenesis and VEGF levels	EGCG single doses 4 h after sunitinib treatment	MCF-7 and H460 xenograft tumors	Downregulation of IRS-1 levels and suppressed mitogenic effects. Marked suppression of the IRS/MAPK/p-S6K1 signaling cascade.	[61]

**Table 2 ijms-24-10737-t002:** EGCG nanomaterials as drug delivery for breast cancer treatment.

Formulation	Study	Results	References
In vitro studies			
Dimeric-EGCG oxidized and polymerized.	Competitive inhibition of Amphiregulin (AREG) in MDA-MB-231 cells.	Proliferation and migration were significantly inhibited by dimeric-EGCG at 10 μM.	[74]
Peracetate-protected (−)-EGCG (Pro-EGCG).	Anticancerogenic effects in MDA-MB-231 tumors.	Enhanced tumor and proteasome inhibition, apoptosis induction, and accumulation.	[75]
Gold nanoparticles (AuNPs) with ratios EGCG/gold 1:2 to 10:1.	Study in MDA-MB-231 cells.	Particles of 39 nm in diameter enhanced irradiation-induced cell death.	[76]
Colloidal mesoporous silica (CMS) and breast tumor-homing cell-penetrating peptide (PEGA-pVEC peptide).	Comparison of anticancerogenic properties of EGCG into CMS and CMS@peptide.	CMS@peptide enhanced the efficacy of EGCG on breast tumors by targeted accumulation and release.	[77]
Specific aptamers to HER2 and ATP organized in a hierarchical manner loaded with EGCG and protamine sulfate.	SK-BR-3; MDA-MB-231.	Improved inhibitory tumor growth and minimum side effects to normal organs and tissues.	[78]
Biodegradable gel: EGCG + siRNA + protamine.	MDA-MB-231 and xenograft MDA-MB-231 tumor-bearing mice.	The formulation enhanced cytotoxicity to cancer cells 15-fold, with little toxicity to normal tissues.	[79]
Nanostructured lipid carriers Arginyl-glycyl-aspartic acid + EGCG; EGCG-loaded NLC-RGD.	Cytotoxic and apoptotic effects and uptake into MDA-MB-231 cells were evaluated.	Nanoparticles with a size of 85 nm enhanced the apoptotic activity of EGCG with higher accumulation in tumors.	[80]
Mesoporous silica gold cluster nanodrug loaded with dual drugs, ZD6474 and EGCG.	Adjuvant treatment to Tamoxifen in MCF-7 and T-47D cells.	The nanoformulation enhanced the toxicity of drugs against chemoresistant cancers.	[81]
2 EGCG nanoparticles FA-NPS-PEG and FA-PEG-NPS.	Modulation of PI3K-Akt pathway and regulatory proteins in MCF-7 cells.	EGCG-FA-NPS-PEG, with a size of 185.0 nm and an encapsulation efficiency of 90.36%, enhanced the cytotoxic activity with IC50 of 65.9 μg/mL.	[82]
FA-NPS-PEG and FA-PEG-NPS nanoparticles.	CNN5 gene activation in MCF-7 (ER-α positive) and MDA-MB-231 (TNBC).	EGCG makes TNBC cells sensitive to estrogen via activating ER-α, reducing the viability and enhancing tumor formation.	[25]
In vivo studies			
EGCG-nanoethosomes, loaded with docetaxel (DT).	Transdermal delivery using mouse skin and treatment of skin cancer growth.	Mice treated with DT-EGCG-nanoethosomes exhibited a significant tumor size reduction by 31.5% after 14 d.	[83]
Natural nanovehicles (exosome-like) from tea flowers (TFENs), particle sizes 131 nm.	Evaluation of tumor growth and metastasis.	Inhibition of growth and tumor metastasis.	[84]
Encapsulation of EGCG in ultradeformable colloidal vesicular systems or penetration enhancer-containing vesicles (PEVs).	Study of photodegradation, stability, and anticancer properties.	EGCG-loaded PEVs increase the cytotoxic activity of epidermoid carcinoma cells (A431) and reduce tumor sizes.	[85]
PC@DOX-PA/EGCG nanoparticles: Phosphatidylcholine, doxorubicin, and procyanidin with HER2, ER, and PR ligands on the surface.	Antitumor evaluation activity in BT-474, MCF-7, EMT-6, and MDA-MB-231.	Nanoparticles can target breast cancer cells and inhibit tumoral growth.	[86]
Folate peptide nanoparticles loaded with EGCG (FP-EGCG-NPs).	Antitumor activity in MDA-MB-231 and MCF-7 cells.	FP-EGCG-NPs enhanced the antitumor activity.	[87]
EGCG in solid lipid nanoparticles conjugated to gastrin-releasing peptide receptors (GRPR).	Tumoral studies on C57/BL6 mice.	Enhanced cytotoxicity to cancer cells, reduction in tumor volume, and greater animal survivability.	[88]

## Data Availability

Not applicable.

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
