# Peer review of "The Potential Role of Epigallocatechin-3-Gallate (EGCG) in Breast Cancer Treatment"

_ijms, 2023, doi:10.3390/ijms241310737_

Round 1
Reviewer 1 Report
The manuscript by Marin et al., is a review article on the impact of EGCG in breast cancer. There are several issues with this article.
1. Table 3 is completely irrelevant in the context of breast cancer. No clinical trial in EGCG+breast cancer?!
2. For table 1 and 2, the authors should provide a forest plot based on the stringency of each study and level of significance.
3. Several sentences are misleading or unclear. For eg: the first sentence in abstract: “Breast cancer is the first diagnosed worldwide with….”; Breast cancer is neither the first diagnosed cancer nor it is the most treatable cancer.
4. EGCG role in tumor immunology is a very weak write up. The details should be provide on the molecular mechanisms of EGCG on individual tumor immune cells like macrophages, CD4+, MDSC etc.
5. In section2: several divergent pathways are presented without any thread tying them together. This molecular details should be presented in the context of the seven hallmarks of carcinogenesis, and the role of EGCG on each of the hallmark.
Author Response
Dear Reviewer,
We thank your corrections and the time used on this. We rewrote many parts of the original document following your suggestions, here we attached the new manuscript.
all the best,
Dr. Cristian Paz
Reviewer 2 Report
The article by Víctor Marín et al. entitled " The potential role of epigallocatechin-3-gallate (EGCG) in breast cancer treatment" is quite interesting. However, it still raises the following issue.
1. The list of contents and subtitles does not support the hypothesis. Hence, authors can revise the manuscript according to the literature contained in vitro, in vivo, and clinical evidence substantiating the anti-breast cancer effects of ECGC and its molecular mechanisms through tables and figures.
2. The tables need to be prepared in vitro and in vivo separately and provide details of the dose, route, and duration of the treatment, various breast cancer cell types, xenograft animal models, outcomes of the study, mechanisms, and references.
3. Manuscripts can be written as anti-breast cancer effects of ECGC in controlling breast cancer proliferation, invasion, migration, metastasis, inflammation, and promoting apoptosis, and epigenetic modification using signaling pathways
Author Response

(The authors gave the same response as above.)

Round 2
Reviewer 2 Report
Accept in present form
Author Response
Dear reviewer,
we checked carefully the manuscript and some misspellings and sentences were corrected, I hope that this new version will be better received by you and a broad audience.
Thank you very much.
Dr. Cristian Paz